# Information sharing and channel encroachment in biomass supply chains

Xin Wu[1], Peng Liu[2], Jin Li[1], Jing Gao[1], Guangyin Xu[1,3]*

1 College of Mechanical and Electrical Engineering, Henan Agricultural University, Zhengzhou, Henan Province, China, 2 School of Management Engineering, Henan University of Engineering, Zhengzhou, Henan Province, China, 3 Institute of Agricultural Engineering, Huanghe S &T University, Zhengzhou, Henan Province, China

* xgy4175@henau.edu.cn

**Data Availability Statement:** All relevant data are within the manuscript and its Supporting Information files.

**Funding:** The author(s) received no specific funding for this work.

## Abstract

To guarantee the sustainable development of the biomass raw material supply chain, researchers are increasingly focusing on the issue of information asymmetry between biomass power plants and upstream supply chain members. This paper investigates the optimal information sharing strategy for a biomass power plant where farmers choose whether to encroach on the biomass feedstock supply. Using a game theory model, we analyze eight different information sharing scenarios, and the results show that when the encroachment occurs in supply chain channels, information sharing can significantly increase the profits of the entire supply chain. In this case, the power plant should share its demand information with all upstream players to promote the overall benefit of the supply chain. In contrast, when the power plant shares its information only with the middleman, it can maximize its profits, which, however, may not be conducive to the long-term stability of the supply chain. Furthermore, surprisingly, in the absence of channel encroachment, the power plant sharing information with upstream members may harm their profits. This suggests that power plants may need to consider the scope of information sharing more carefully when the farmers choose not to encroach. Finally, we also examine the impact of channel competition intensity on information sharing strategies, and find that when channel competition intensity is low, transparent demand information helps the power plant maximize expected returns. However, in a highly competitive market environment, the power plant should carefully handle information sharing with farmers to avoid damaging their profits.

## 1. Introduction

As the global demand for renewable energy continues to grow, biomass power generation has become an essential contributor to the global energy transition as a key means to achieve energy security and rural development [1]. Since the 1970s, biomass power generation technologies have significantly developed in many European and American countries. The utilization of biomass energy not only reduces reliance on fossil fuels, ensures energy security and mitigates environmental pollution [2]. Especially in China, the technological progress and

**Competing interests:** The authors have declared that no competing interests exist.

environmental benefits of biomass power generation are driven by policies, showing significant emission reduction potential. A comprehensive life cycle assessment of Chinese data indicates that integrating biomass pyrolysis and power generation with common methane and nitrogen reduction measures can significantly reduce the greenhouse gas emissions of significant crops from 666.5 TgCO2 equivalent per year to -37.9 TgCO2 equivalent [3]. China has committed to peaking carbon emissions by 2030, achieving carbon neutrality by 2060, and increasing the share of non-fossil fuels in primary energy consumption to 20% [4]. Therefore, reducing fossil fuel consumption and replacing it with renewable energy is fundamental to achieving carbon neutrality [5]. However, supply chain inefficiencies in biomass power generation, particularly in feedstock supply and information sharing, limit the realization of this potential. This study aims to explore the effectiveness of information sharing strategies in biomass supply chains under channel encroachment and propose strategies to promote the sustainable development of biomass energy supply chains.

Although the Chinese government has been encouraging the development of biomass power generation in recent years, there is still considerable room for the development of biomass power generation [6]. In 2022, China's total biomass power generation witnessed a notable increase of 11.42% to reach 182.4 billion kWh, as reported by the National Energy Administration (NEA). Despite this growth, there was a decline of 1.7% in the proportion of biomass power generation within the overall power generation structure compared to the previous year. Among these, the proportion of electricity generation from municipal solid waste incineration increased from 52% in 2020 to 70% in 2022, while the proportion of electricity generation from agricultural and forestry biomass decreased from 45% in 2020 to 28% in 2022 [7]. In biomass straw power plants, one obstacle to bioenergy development is the difficulty in maintaining the sustainability of the biomass power supply chain [8]. Among the existing biomass feedstock supply models, the "farmer → broker → power plant" model is the most common straw feedstock acquisition model in China [9]. Under this model, middlemen control the feedstock supply, resulting in high supply costs. In reality, it is also common for farmers to sign contracts with biomass power plants to supply biomass feedstock. This model clearly creates channel competition with the traditional indirect supply model of middleman. Wen and Zhang [10] verified that a hybrid acquisition model combining simultaneous direct purchase from farmers by biomass power plants and straw purchase through an middleman can achieve biomass feedstock supply at a lower cost.

On the other hand, material demand forecasting in the supply chain is also an important tool to improve the continuous operation of the grid. Precise and rational material demand predictions serve as a solid basis for material procurement, enhancing proactive enterprise material management practices and facilitating advanced resource planning within enterprises. For example, in Dehui City, Jilin Province, straw fuel utilization for power generation is mainly through direct combustion. Biomass Power Generation (Dehui) Co., Ltd. and State Power Dehui Biomass Power Generation Co., Ltd. have actively communicated and conducted on-site inspections to grasp the current raw material reserves fully and demand, coordinating agreements between enterprises and farmers or cooperatives to ensure the supply of straw. Material demand management in power generation enterprises relies heavily on energy demand planning at the grassroots level in different cities, in addition to field surveys. The head of each supply department conducts statistical forecasts by estimating material demand for the upcoming quarter or year within their respective areas of responsibility. These individual estimates are aggregated and analyzed to project the overall material demand for the company in the upcoming quarter or year. It is important to note that this approach may be influenced by subjective factors at the individual level [11]. Additionally, He [12] utilized the Long-range Energy Alternatives Planning system (LEAP) to construct a planning model for

the electricity industry, aiming to offer a more scientifically grounded reference for biomass power generation planning in China. Other researchers have forecasted the raw material demand for biomass power plants by considering the heat and power generation of such plants as well as per capita energy consumption [13]. In practice, most biomass power plants predict raw material demand based on local power generation plans, boiler capacity, and operating time of power generation projects. For example, Zhecheng County, Shangqiu City, Henan Province's biomass cogeneration project is equipped with a 1×130t/h high-temperature and high-pressure straw boiler. It consumes about 219,300 tons of fuel annually at rated load. Whether biomass power plants can reasonably use their demand forecasting information is crucial for the sustainable supply of biomass straw [14].

According to existing research, information sharing has been recognized as an effective approach to improving overall supply chain performance. Information sharing not only helps reduce the risks caused by information asymmetry in the supply chain, but also enhances the stability of the supply chain [15].For example, in the comprehensive utilization of straw in Jiaxian County, Henan Province, the government and enterprises take the lead in building a straw collection and storage information platform, so that the utilization rate of straw in the county has reached more than 80% [16]. In order to promote crop straw storage and comprehensive utilization, Guiyang Municipal Bureau of Agriculture and Rural Development organized the city's straw storage and utilization of the main demand for information and to be announced on the official website [17]. Although there are examples of biomass supply chain information platforms, information asymmetry is still a common problem. Due to information asymmetry between upstream and downstream not only hinders the continuous production of power plants but also dampens farmers' enthusiasm as the supply source [18]. Since most biomass power plants purchase straw through middlemen, farmers need to be aware of the specific demand information of biomass power plants, leading middlemen to profit from information asymmetry [19]. However, the cost of collecting biomass raw materials is mainly borne by farmers, and the lack of labor and meager profits make farmers unwilling to collect straw in the fields [20]. Instead, they may burn straw directly before spring sowing to avoid processing costs [21]. Zhai et al. [22] focused on the vertical interaction between farmers, biomass power plants, and the government in the biomass power generation supply chain, discussing the government's policy choice based on asymmetric information and factors affecting Nash equilibrium. Evidently, the asymmetry of information in the biomass supply chain also hinders policymakers and planners from formulating more practical sustainable energy policies [23]. In 2022, the National Development and Reform Commission (NDRC)'s guidance for green and low-carbon energy transformation also explicitly stated the need to build an energy basic information platform and integrate energy supply chain information [24]. Therefore, resolving biomass system failures will require detailed information about biomass consumption and supply, to be shared in the supply chain, thereby achieving the sustainable development of the biomass supply chain. While ensuring the interests of all parties, how to effectively share demand information to curb biomass resource waste, increase farmers' enthusiasm to supply biomass straw to power plants, and achieve the sustainable development of the biomass power generation supply chain is a worthy research problem.

Based on the above, we investigate the information sharing strategies of biomass power plants in the biomass power supply chain in the presence or absence of channel encroachment, which contains the following questions:(1) What are the equilibrium outcomes under different information sharing strategies depending on whether farmers choose channel encroachment or not? (2) How do information sharing strategies of biomass power plants affect the price decisions of middlemen and farmers regarding straw? (3) Do biomass power plants have the motivation to share their demand information voluntarily? If so, which information sharing

strategy is optimal for the supply chain? Which information sharing strategy do other members of the supply chain tend to prefer?

Given these questions, we studied a biomass supply chain consisting of power plants, middlemen, and farmers, reflecting the actual scenario of a biomass power generation supply chain: farmers selling available biomass to middlemen, who collect and sell it to power plants. Farmers can also directly collect and sell biomass straw to power plants. Biomass power plants can predict demand information based on current raw material reserves and the configuration of biomass power generation units, coordinate agreements with agricultural entities such as farmers and middlemen through on-site inspections to ensure their demand supply, and decide whether to share this private information with upstream supply chain members. Specifically, we compared eight information sharing scenarios regarding whether farmers choose to encroach on the middleman channel and how power plants conduct their information sharing strategies.

The results are as follows: Firstly, farmers are willing to broaden their channels for selling biomass raw materials to increase their profits. Secondly, the equilibrium prices of supply chain members are influenced by the information sharing strategies of power plants and farmers' choice of encroachment strategy. Both farmers and middlemen want exclusive access to demand information from biomass power plants to maximize their profits. If farmers choose to encroach on biomass supply channel, it will correspondingly reduce the expected wholesale price of biomass raw materials from farmers by middlemen. However, when the demand information forecast by power plants is high, and middlemen exclusively share demand information, the encroachment of farmers has no significant impact on the unit selling price of biomass raw materials from farmers to middlemen. Furthermore, we obtained the optimal information sharing strategy for biomass power plants. Power plants should only share demand information with middlemen in non-encroachment scenarios to maximize their expected profits. In encroachment scenarios, biomass power plants should actively predict demand information and try to make the information transparent. However, when the competition intensity in the channel is too high, power plants should conceal demand information to reduce the risk of price wars.

This study makes several contributions to the literature as follows. Firstly, it focuses on the information sharing strategy of a biomass supply chain consisting of a power plant, an middleman, and farmers. This supply chain structure is common in biomass energy supply but lacks related research on information sharing strategies. Secondly, we derive pricing equilibrium strategies for different information sharing scenarios, using whether farmers choose the encroachment channel as an entry point. Thirdly, we further compared the expected profits of biomass supply chain members in different channel information scenarios. The study provides several practical contributions. It demonstrates how farmers' encroachment strategies and power plants' information sharing strategies influence the equilibrium prices of supply chain members under different circumstances. This study also suggests that when channel competition is intense, sharing demand information by biomass power plants may put them at a disadvantage. Moreover, we further illustrate how biomass power plants should disclose or conceal their quality information in the case of information asymmetry to maximize their expected returns.

## 2. Literature review

This paper focuses on the impact of information sharing strategies on the pricing decisions of supply chain members in the presence of channel encroachment, which is related to three

research areas in the literature focused on biomass supply chains, channel encroachment and information sharing strategies.

Memişoğlu and Üster [25] state that bioenergy is a new area of study that has caught the interest of supply chain management experts. The stream of literature on the bioenergy supply chain policies and subsidies has been particularly extensive. For example, Ma [26] synthesized existing research regarding the policy frameworks and subsidies to support the biomass industry and employed computer simulation to analyze the forest biomass power generation supply chain system, discussing the impact of various forestry policies on the chain's profitability. The literature indicated that optimizing operational efficiency and strategic information sharing strategies can improve overall supply chain sustainability and performance. Jiang et al. [20] highlighted subsidies as an important strategic tool used by governments to promote renewable energy and ensure its integration into the energy mix, and researched optimal subsidy strategies by the government for the biomass raw material supply chain composed of power plants, village committees, and farmers. Their analysis revealed the optimal decision strategies for the stakeholders involved. This provides a basis for exploring the strategic role of information sharing in biomass supply chains. Li et al. [27] consolidated research on policies such as feed-in tariffs and their impact on the biomass supply chain and analyzed the relationship between government funding and the on-grid electricity price of biomass power plants, demonstrating that when government funding is below a certain threshold, technical subsidies should be preferred over on-grid electricity price subsidies. The study provides a foundation for the nuanced information sharing strategies needed to analyze channel encroachment into the biomass power industry. Alongside the extensive examination of subsidy policies, a parallel discourse exists on the interplay between these policies and supply chain risks. For instance, Liu et al. [28] investigated the current challenges facing the biomass power generation industry. They proposed that the government implement supportive policies at various supply chain stages. Additionally, research has delved into the connection between policies and supply chain risks. Furthermore, some scholars' conversations extend into biomass supply chain optimization and network design. Here, researchers concentrate on developing models that enhance the biomass supply chain's efficiency while considering various logistical and environmental restrictions. Pérez-Fortes et al. [29] constructed a multi-objective mixed-integer linear programming model, revealing the trade-offs between economic, environmental, and social standards. Shabani and Sowlati [30] established a non-linear mixed-integer programming model to manage and optimize the forest biomass supply chain, evaluating different schemes for maximizing biomass availability and investing in new ash recovery systems. Zhao and Li [31] focused on logistics cost optimization and minimizing pollutant emissions, establishing a multi-objective integer programming model for the optimal siting of biomass power plants and corresponding raw material supply chain design. Their study demonstrated the model's ability to balance environmental and economic benefits. In contrast to the above literature, our work is centered on optimal information sharing strategies under asymmetric information. Importantly, we introduce the potential impact of farmers encroaching biomass supply channels. We are interested in understanding how power plants' information sharing strategies affect supply chain members' interests under asymmetric information. This study centers on the decision-making processes of stakeholders, including biomass power plants, biomass feedstock middlemen, and farmers, across various scenarios. Game theory offers a valuable framework for comprehending stakeholder interactions in the context of optimizing the biomass supply chain. Wang and Watanabe [32] examined the incentive effects in China's supply chain for straw-based power plants under perceived risks using a Stackelberg game. They developed incentive scenarios grounded on stakeholders' risk perceptions to boost the supply of straw feedstock and enhance the profitability of the supply chain. The study offers

crucial insights into the strategic behaviors and decision-making processes that emerge in response to risks and policy invention in the biomass power industry. Zhang et al. [33] Literature used game theory to study the strategic interactions between different stakeholders and emphasized the complexity faced in formulating information-sharing strategies within the biomass power generation supply chain. There is complexity in formulating information-sharing strategies within the chain, especially in the case of channel encroachment. They considered the impact of the non-material utility of stakeholders in the biomass power supply chain and the formal organization of farmers on farmer behavior. This work lays the foundation for this study, providing a comprehensive context that facilitates specific research on information sharing strategies. Most studies on biomass energy supply chains generally assume symmetric information. In practice, there are frequently "information silos" that exist among participants within the biomass supply chain. Our work contributes to this literature by studying the information sharing strategies of biomass power plants under asymmetric information, providing valuable insights. Unlike existing literature, we assume, based on practical scenarios, that in the biomass energy supply chain, the demand information for biomass raw materials is initially forecasted by the power plant and is private to the biomass power plant. This information is unobservable to the suppliers of biomass raw materials, namely, farmers and middlemen. The study results describe the impact of asymmetric demand information on the equilibrium prices of various parties in the biomass raw material supply chain and how biomass power plants should set optimal information sharing strategies under asymmetric information.

As mentioned before, channel encroachment is growing and increasingly popular. The concept relates to scenarios where manufacturers or primary suppliers sell directly to end customers instead of through traditional distribution channels, possibly disrupting established supply chains [34]. It requires investigating the strategies that biomass supply networks use to mitigate the risks and leverage the opportunities presented by such direct engagements. Some literature mentions the raw material supply issue in biomass supply chains. Most scholars advocate reducing the uncertainty and cost of biomass supply systems by using comprehensive multi-feedstock bioenergy (i.e., biofuels, bioenergy, or bioproducts) supply systems [35–38]. However, in addition to improving supply chain efficiency, supply chain cooperation mechanisms also need to be considered. In this context, the study of Cai [39] provides insights from another perspective. Different from the above studies that mainly focus on supply chain efficiency. The study explored two raw material supply models in the bioenergy supply chain: Contract Farming (CF) and Land-set Equity (LS). Cai's research found that these cooperation models can promote cooperation between farmers and biomass producers and effectively improve the supply capacity of biomass raw materials. Yuan et al. [40] used linguistic hesitant fuzzy sets to evaluate biomass power generation fuel procurement and storage modes, proving that the "farmer + middleman + third-party logistics + power plant" mode is the most reasonable for the development of biomass energy in Jilin Province. Based on Cai's [39] research on cooperation models, a more in-depth exploration of biomass supply channels has become the focus of research. For example, Tan et al. [41] used factor analysis to identify the main factors affecting farmer willingness and behavior and proposed new channels conducive to biomass harvesting. Key factors influencing farmer participation in bioenergy supply chains are revealed. These factors include but are not limited to, financial incentives, environmental awareness, and policy support. Their findings indicate that while multiple channels exist to facilitate biomass supply, farmers' individual decisions have a decisive impact on the actual amount of biomass supplied. This emphasizes the need to take farmers' wishes and behavioral patterns into account when designing supply chain strategies to ensure the effectiveness of the strategy and the sustainability of the supply chain. This echoes the research of Tan et al. [41], Luo et al. [42], Fan et al. [43] and Sun et al. [44], all of whom pointed out that optimizing supply channels and

mobilizing the enthusiasm of farmers are crucial to deal with the challenge of channel encroachment. Luo et al. [42] introduced rural formal organizations to broaden channels, finding that the cooperative enthusiasm of rural official organizations positively impacts farmer participation. At the same time, profit-sharing policies have a significant impact on the balance ratio of cooperative farmers. Fan et al. [43] constructed a biomass supply chain model consisting of farmers, middlemen, and manufacturers. Farmers decide whether to supply manufacturers or middlemen, and middlemen decide to supply manufacturers. The study achieved supply chain coordination by designing "protective price + subsidy" contracts and "repurchase + profit-sharing" contracts. Sun et al. [44] researched the optimal strategies of a biomass supply chain with buyer channel competition, building a game model consisting of one supplier and two buyers. The above research highlights the impact of biological raw material supply channels on supply chain stability, pointing out the importance of farmers' willingness, optimization of supply channels, and the role of rural official organizations and other factors. However, this literature series usually does not fully consider the coordination effect of biomass power plants on the supply chain as the main demander of biomass raw materials when discussing supply chain management. In actual operations, power plants often predict and calculate the demand for biomass raw materials. This demand-pull mechanism promotes the response accuracy of upstream raw material supply while reducing redundant inventory costs in the supply chain [45]. Against this background, our research focuses on the impact of demand information on supply chain members' decisions in a biomass feedstock supply chain. By in-depth analysis of how power plants influence supply chain dynamics and decisions, we not only reveal a new perspective on supply chain coordination but also provide helpful management implications for the development of biomass feedstock supply chains, especially on how to effectively integrate demand information to Improve the overall performance of the supply chain.

In reality, the operational decision-making of supply chain upstream and downstream businesses is significantly impacted by information asymmetry [46, 47]. Examining information sharing strategies is vital in addressing the disparities in accessible data between upstream and downstream businesses, enabling more synchronized and efficient decision-making processes across the entire supply chain [48]. On this basis, some scholars have further conducted research on information sharing strategies of different supply chain members. Yoon et al. [49] found that information sharing makes manufacturers' procurement decisions more conservative, potentially leading to an increase in raw material inventory. The procurement decisions of manufacturing companies depend on the reliability of suppliers; when suppliers are unreliable (reliable), manufacturing companies will correspondingly be more proactive (conservative), carrying fewer (more) inventories. The effectiveness of sharing information depends on various factors, including the reliability of suppliers and information sharing costs. Wang et al. [50] demonstrated that middlemen always have a voluntary motivation to share information, and their optimal strategy depends mainly on channel competition intensity and related costs. When manufacturers cooperate with retailers, middlemen always intend to share information, whereas if middlemen and retailers form an alliance, there is no motivation to share information. In this context, the asymmetry of demand information becomes our focus. Some scholars have examined how when demand information asymmetry exists in the supply chain; this imbalance affects the decision-making process of supply chain members. Especially in a complex network such as the biomass supply chain, accurate demand information is crucial to upstream producers, directly affecting their production plans and resource allocation. For example, Li and Zhang [51] studied the relationship between the motivation for demand information sharing and the dependence on manufacturers' operations and decisions. When the degree of demand uncertainty is moderate, retailers are motivated to share information

**Table 1. Comparison with the literature.**

| | Biomass supply chain | | | Channel | | Information | | | |
|---|---|---|---|---|---|---|---|---|---|
| | Behavioral decision | Logistics network | Policy and risk control | Encroachment | Non-encroachment | Symmetric | Asymmetric | Shared | Unshared |
| [25] | | ✓ | ✓ | | | | | | |
| [26] | ✓ | | ✓ | | | | | | |
| [27] | ✓ | | ✓ | ✓ | ✓ | ✓ | | | |
| [28] | | | ✓ | | | | | | |
| [29] | | ✓ | | | | | | | |
| [30] | ✓ | | | | | | ✓ | | |
| [31] | | ✓ | | | | | | | |
| [20] | ✓ | | ✓ | | | | ✓ | ✓ | ✓ |
| [32] | ✓ | | ✓ | | ✓ | ✓ | | | |
| [33] | ✓ | | | ✓ | | ✓ | | | |
| [34, 46, 47, 50, 52] | | | | ✓ | ✓ | | ✓ | ✓ | ✓ |
| [35–38] | | ✓ | | | | | | | |
| [39] | ✓ | | ✓ | | | | | | |
| [40] | ✓ | | | | ✓ | | ✓ | | |
| [41, 44] | ✓ | | | ✓ | | | | | |
| [42, 43] | ✓ | | ✓ | ✓ | | | | | |
| [48, 51] | | | | | | | ✓ | ✓ | ✓ |
| This paper | ✓ | | | ✓ | ✓ | | ✓ | ✓ | ✓ |

voluntarily with manufacturers operating on inventory production. Retailers have no incentive to share information when manufacturers operate on order production. Yu et al. [52] analyzed the impact of traditional supply chain demand information sharing under carbon constraints and transaction regulations. The results indicated that when demand uncertainty is relatively low, information sharing is beneficial to existing retailers. Manufacturers always benefit from the demand for information sharing by retailers and should strive to improve the benefits of information sharing for carbon reduction. These studies all indicate that information sharing significantly impacts supply chain management. However, the impact of information sharing on the biomass raw material supply chain has yet to be studied in the literature. Our research further delves into the role that demand information asymmetry plays in supply chains and how to strengthen cooperation among supply chain members by improving information-sharing mechanisms, thereby improving the entire system's efficiency. Unlike existing literature, this paper introduces the encroachment of farmers into the biomass raw material supply chain. It compares the optimal decisions of power plants under single and multiple biomass raw material supply channels. The results of the study reveal which demand information sharing strategy can increase the profits of biomass supply chain participants in different scenarios. The comparison of our paper with the extant literature is shown in Table 1.

## 3. Model framework

We consider a biomass supply chain composed of a group of farmers (F), an middleman (M), and a power plant (P). The farmers collect biomass and sell it to the middleman, who, in turn, sells it to the power plant for electricity generation. Farmers also have the opportunity to encroach by selling directly to the power plant. The power plant possesses relevant data and can forecast the future demand for biomass raw materials. It decides whether and to which

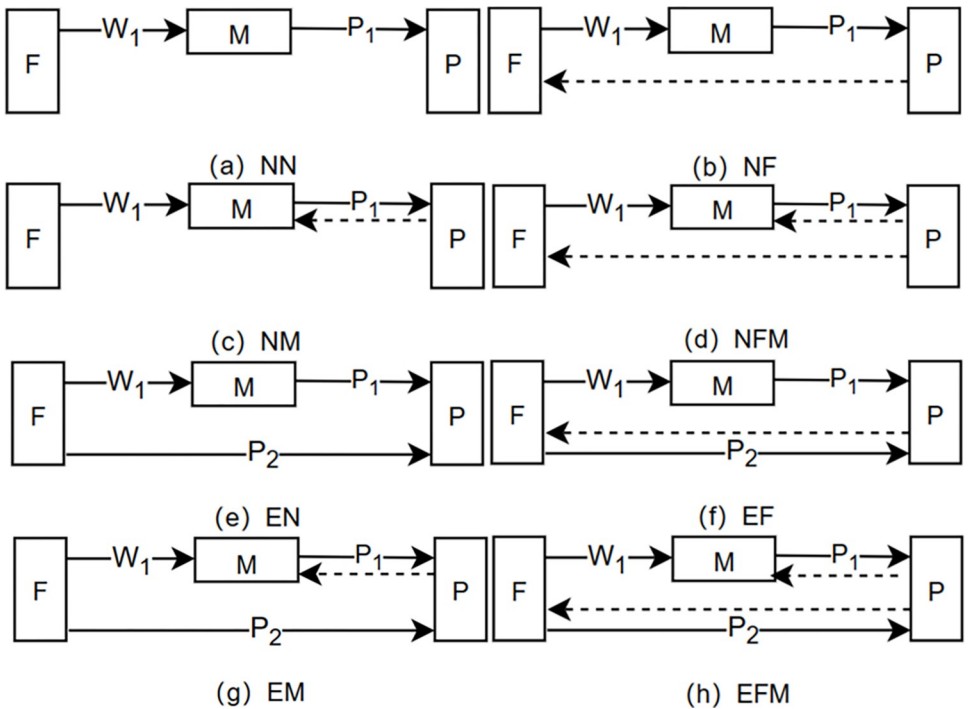

**Fig 1. Eight demand information sharing scenarios for the biomass power supply chain.** Farmers do not choose channel encroachment, and the power plant does not share demand information (Scenario NN, Figure a); Farmers do not choose channel encroachment, and the power plant shares information with farmers (Scenario NF, Figure b); Farmers do not choose channel encroachment, and the power plant shares information with the middleman (Scenario NM, Figure c); Farmers do not choose channel encroachment, and the power plant shares information with both farmers and the middleman (Scenario NFM, Figure d); Farmers choose channel encroachment, and the power plant does not share demand information (Scenario EN, Figure e); Farmers choose channel encroachment, and the power plant shares information with farmers (Scenario EF, Figure f); Farmers choose channel encroachment, and the power plant shares information only with the middleman (Scenario EM, Figure g); Farmers choose channel encroachment, and the power plant shares information with both farmers and the middleman (Scenario EFM, Figure h).

upstream supply chain member to share this demand information. Based on the scenarios of farmer encroachment (N, E) and power plant information sharing strategy (N, M, F, MF), there are a total of eight situations (Fig 1). Furthermore, similar to Jiang et al. [53], we assume that the unit costs for farmers, middlemen, and power plants in handling this biomass are denoted as $c_0/c_1/c_2/c_3$ in different channels.

Similar to Yan et al. [54] and Li et al. [55], the biomass plant demand is assumed to be a downward sloping linear curve of the following form. When farmers do not encroach, the demand function is $d_1 = a-p_1$. When farmers encroach, the demand functions are $d_1 = a -p_1+bp_2$ and $d_2 = a-p_2+bp_1$. Here, $d_i$ and $p_i$ are the demand quantity and unit price for selling biomass to the power plant in the channel $i$ (where $i = 1$ and 2 represent the middleman and farmer encroachment channels, respectively), and a represents the potential demand. The parameter $b(1>b>0)$ indicates the intensity of competition between the two channels.

Based on the real scenario of the biomass power generation supply chain, let's assume that the biomass raw material demand is stochastic. Specifically, let the uncertain demand scale be represented by $= a_0 +\vartheta$, where $a_0$ is the fixed demand and $\vartheta$ represents the fluctuating demand with mean 0 and variance v. Biomass power plants forecast future demand based on actual situations to obtain market signals $f$. However, due to the inability of demand forecast data to be equal to real demand, errors inevitably exist. The error term is denoted by $\varepsilon$ (i.e., $f = a+\varepsilon$),

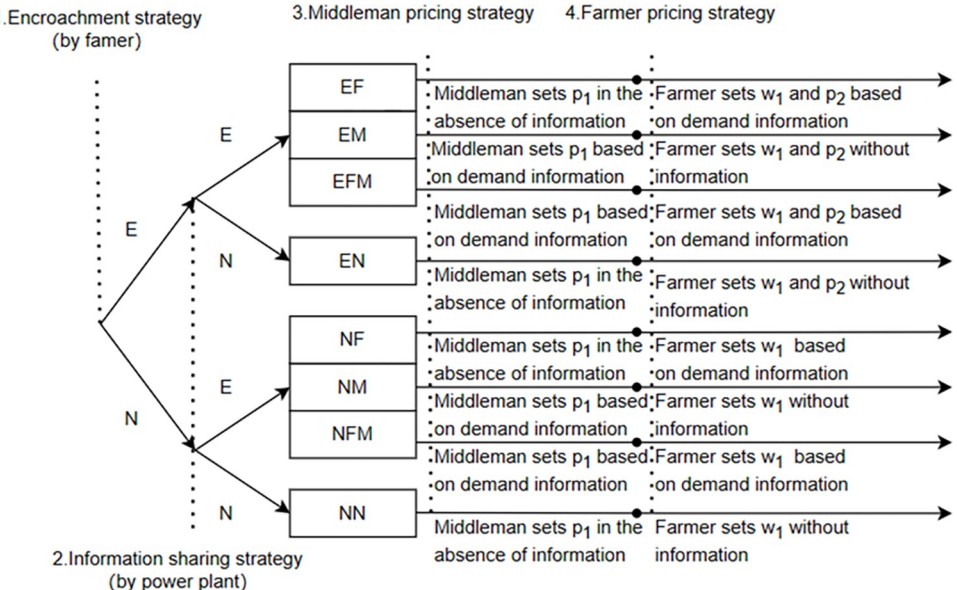

**Fig 2. Sequence of events.**

where $\varepsilon$ is a random variable with mean zero and variance s. Similar to Liu et al. [56], suppose the parameters $\varepsilon$ and $\vartheta$ are independent, from which the information structure can be derived:

$$E(a|f) = \frac{s}{s+v}a_0 + \frac{v}{s+v}f \equiv A \tag{1}$$

$$E((f-a_0)^2|f) = s+v \tag{2}$$

The expression $E(a|f)$ When representing the market signal as $f$, the conditional expected market size is. $t = \frac{v}{s+v}, (0 < t < 1)$ represents the accuracy of demand forecasting, when $t = 1$, maximum accuracy of demand forecasting, in other words, $A = a$. In contrast, the lowest accuracy of prediction occurs when $t = 0$, representing a scenario entirely different from the actual market signal.

The biomass power supply chain is essentially a supply chain primarily driven by production (electricity generation). The encroachment decision is assumed to be made before the sharing strategy because, in general, the power plant holds a leadership position in the supply chain, and after observing the upstream structure, the power plant decides whether to share information with upstream partners. Additionally, similar to Wei et al. [57], this chapter sets the corresponding pricing decision sequence. As shown in Fig 2, the decision sequence is as follows: (1) Farmers decide whether to encroach; (2) After receiving the market signal $f$, the power plant decides whether to share information upstream; (3) The middleman first sets the unit price $p_1$ for selling biomass to the power plant; (4) Farmers set the wholesale price $w_1$ for selling biomass to the middleman and the unit price $p_2$ for selling biomass directly to the power plant. This sequence illustrates the game relationships among farmers, middleman, and the power plant.

Next, we will derive equilibrium wholesale and unit selling prices for these eight subgames and calculate the equilibrium profit of supply chain members. Finally, we will explore the interaction between encroachment and information sharing strategies. For clarity, Table 2 lists the parameter symbols and explanations.

**Table 2. Symbols.**

| Symbols | Connotation |
|---|---|
| $w_1$ | Wholesale price |
| $p_1, p_2$ | Unit selling prices in both channels |
| $d_1, d_2$ | Demand from both channels |
| $d_1^N$ | Demand for channel 1 at the time of the no-encroachment |
| $d_1^E$ | Demand for channel 1 at the time of the encroachment |
| $c_0, c_1, c_2, c_3$ | Transport costs at various points, $c>0$ |
| $b$ | Intensity of competition between channels, $1>b>0$ |
| $a$ | Latent market demand |
| $a_0$ | Fixed portion of potential market demand |
| $f$ | Market signal |
| $\vartheta$ | Variable portion of potential demand |
| $v$ | Variance of uncertainty component |
| $\varepsilon$ | Error component of market signals |
| $s$ | Variance of the error component |
| $A$ | Forecasting demand |

# 4. Information sharing scenarios

This section provides pricing equilibrium strategies for each of the eight scenarios.

## 4.1 Scenario NN

In Scenario NN, the farmers wholesale the biomass raw materials to the middleman, who then sells the biomass raw materials to the power plant. Additionally, the power plant does not provide quantity demand information upstream. Therefore, in this scenario, the profit functions for each supply chain member are as follows:

$$E(\pi_P|f) = (p_1 - c_3)d_1^N \tag{3}$$

$$E(\pi_M) = (p_1 - w_1 - c_2)d_1^N \tag{4}$$

$$E(\pi_F) = (w_1 - c_1)d_1^N \tag{5}$$

By utilizing the method of reverse solution, first, differentiate (5) with respect to $w_1$, we obtain the reaction function $w_1 = a_0 + c_1 - p_1$. Substituting this reaction function into (4) and then differentiating (4) with respect to $p_1$, we can obtain $p_1 = \frac{1}{4}(3a_0 + c_1 + c_2)$. Substituting $p_1 = \frac{1}{4}(3a_0 + c_1 + c_2)$ into the aforementioned $w_1 = a_0 + c_1 - p_1$ yields the equilibrium price.

## 4.2 Scenario NF

In scenario NF, the farmers wholesale the biomass to the middleman, which then sells the materials to the power plant; and the power plant provides electricity demand information to the upstream farmers. Therefore, in this scenario, the profit functions of each supply chain member are:

$$E(\pi_P|f) = (p_1 - c_3)d_1^N \tag{6}$$

$$E(\pi_M) = (p_1 - w_1 - c_2)d_1^N \tag{7}$$

$$E(\pi_F|f) = (w_1 - c_1)d_1^N \tag{8}$$

By employing the reverse solution method: first, differentiate (8) with respect to $w_1$, we obtain the reaction function $w_1 = A + c_1 - p_1$. Substituting this reaction function into (7) and then differentiating (7) with respect to $p_1$, we can obtain $p_1 = \frac{1}{4}(3a_0 + c_1 + c_2)$. Substituting $p_1 = \frac{1}{4}(3a_0 + c_1 + c_2)$ into the aforementioned $w_1 = A + c_1 - p_1$ yields the equilibrium price.

### 4.3 Scenario NM

In scenario NM, the farmers wholesale the biomass to the middleman, which then sells the materials to the power plant; and the power plant provides electricity demand information to the upstream middleman. Therefore, in this scenario, the profit functions of each supply chain member are:

$$E(\pi_P|f) = (p_1 - c_3)d_1^N \tag{9}$$

$$E(\pi_M|f) = (p_1 - w_1 - c_2)d_1^N \tag{10}$$

$$E(\pi_F) = (w_1 - c_1)d_1^N \tag{11}$$

By employing the reverse solution method: first, differentiate (11) with respect to $w_1$, we obtain the reaction function $w_1 = a_0 + c_1 - p_1$. Substituting this reaction function into (10) and then differentiating (10) with respect to $p_1$, we can obtain $p_1 = \frac{1}{4}(2A + a_0 + c_1 + c_2)$. Substituting $p_1 = \frac{1}{4}(2A + a_0 + c_1 + c_2)$ into the aforementioned $w_1 = a_0 + c_1 p_1$ yields the equilibrium price.

### 4.4 Scenario NFM

In the NFM scenario, the farmers wholesale the biomass to the middleman, which then sells the materials to the power plant, and the power plant provides electricity demand information to both the upstream farmers and middlemen. Therefore, in this scenario, the profit functions of each supply chain member are:

$$E(\pi_P|f) = (p_1 - c_3)d_1^N \tag{12}$$

$$E(\pi_M|f) = (p_1 - w_1 - c_2)d_1^N \tag{13}$$

$$E(\pi_F|f) = (w_1 - c_1)d_1^N \tag{14}$$

By employing the reverse solution method: first differentiate (14) with respect to $w_1$, we obtain the reaction function $w_1 = A + c_1 - p_1$. Substituting this reaction function into (13), and then differentiating (13) with respect to $p_1$, we can obtain $p_1 = \frac{1}{4}(3A + c_1 + c_2)$. Substituting $p_1 = \frac{1}{4}(3A + c_1 + c_2)$ into the aforementioned $w_1 = A + c_1 - p_1$ yields the equilibrium price.

For convenience, the equilibrium pricing outcomes when farmers opt not to encroach the channels are presented in Table 3.

**Table 3. Equilibrium pricing for scenarios under the non-encroachment channel.**

|       | NN | NF | NM | NFM |
|-------|----|----|----|-----|
| $w_1$ | $\frac{1}{4}(a_0 + 3c_1 - c_2)$ | $\frac{1}{4}(4A - 3a_0 + 3c_1 - c_2)$ | $\frac{1}{4}(3a_0 - 2a + 3c_1 - c_2)$ | $\frac{1}{4}(A + 3c_1 - c_2)$ |
| $p_1$ | $\frac{1}{4}(3a_0 + c_1 + c_2)$ | $\frac{1}{4}(3a_0 + c_1 + c_2)$ | $\frac{1}{4}(2A + a_0 + c_1 + c_2)$ | $\frac{1}{4}(3A + c_1 + c_2)$ |

### 4.5 Scenario EN

In scenario EN, the farmers not only sell biomass to the power plant through the middleman but also directly to the power plant; and the power plant does not provide electricity demand information to the upstream. Therefore, in this scenario, the profit functions of each supply chain member are:

$$E(\pi_P|f) = p_1 d_1^E + p_2 d_2 - c_3(d_1^E + d_2) \tag{15}$$

$$E(\pi_M) = (p_1 - w_1 - c_2)d_1^E \tag{16}$$

$$E(\pi_F) = (w_1 - c_1)d_1^E + (p_2 - c_0)d_2 \tag{17}$$

By employing the reverse solution method: first differentiate (17) with respect to $w_1$ and $p_2$, we obtain the reaction functions $w_1 = \frac{a_0 + c_1 - bc_1 - p_1 + bp_1}{1-b}$ and $p_2 = \frac{a_0 + c_0 - bc_0}{2(1-b)}$. Substituting these reaction functions into (16), and then differentiating (16) with respect to $p_1$, we can obtain $p_1 = \frac{-3a_0 + a_0 b - bc_0 + b^2 c_0 - c_1 + bc_1 - c_2 + bc_2}{4(-1+b)}$. Substituting $p_1 = \frac{-3a_0 + a_0 b - bc_0 + b^2 c_0 - c_1 + bc_1 - c_2 + bc_2}{4(-1+b)}$ into the aforementioned $w_1 = \frac{a_0 + c_1 - bc_1 - p_1 + bp_1}{1-b}$ yields the equilibrium price.

### 4.6 Scenario EF

In scenario EF, the farmers not only sell biomass to the power plant through the middleman but also directly to the power plant; and the power plant provides electricity demand information to the upstream farmers. Therefore, in this scenario, the profit functions of each supply chain member are:

$$E(\pi_P|f) = p_1 d_1^E + p_2 d_2 - c_3(d_1^E + d_2) \tag{18}$$

$$E(\pi_M) = (p_1 - w_1 - c_2)d_1^E \tag{19}$$

$$E(\pi_F|f) = (w_1 - c_1)d_1^E + (p_2 - c_0)d_2 \tag{20}$$

By employing the reverse solution method: first differentiate (20) with respect to $w_1$ and $p_2$, we obtain the reaction functions $w_1 = \frac{A + c_1 - bc_1 - p_1 + bp_1}{1-b}$ and $p_2 = \frac{A + c_0 - bc_0}{2(1-b)}$. Substituting these reaction functions into (19), and then differentiating (19) with respect to $p_1$, we can obtain $p_1 = \frac{-3a_0 + a_0 b - bc_0 + b^2 c_0 - c_1 + bc_1 - c_2 + bc_2}{4(-1+b)}$. Substituting $p_1 = \frac{-3a_0 + a_0 b - bc_0 + b^2 c_0 - c_1 + bc_1 - c_2 + bc_2}{4(-1+b)}$ into the aforementioned $w_1 = \frac{A + c_1 - bc_1 - p_1 + bp_1}{1-b}$ yields the equilibrium price.

### 4.7 Scenario EM

In scenario EM, the farmers not only sell biomass to the power plant through the middleman but also directly to the power plant; and the power plant provides electricity demand information to the upstream middleman. Therefore, in this scenario, the profit functions of each

supply chain member are:

$$E(\pi_P|f) = p_1 d_1^E + p_2 d_2 - c_3(d_1^E + d_2) \tag{21}$$

$$E(\pi_M|f) = (p_1 - w_1 - c_2)d_1^E \tag{22}$$

$$E(\pi_F|f) = (w_1 - c_1)d_1^E + (p_2 - c_0)d_2 \tag{23}$$

By employing the reverse solution method: first differentiate (23) with respect to $w_1$ and $p_2$, we obtain the reaction functions $w_1 = \frac{a_0 + c_1 - bc_1 - p_1 + bp_1}{1-b}$ and $p_2 = \frac{a_0 + c_0 - bc_0}{2(1-b)}$. Substituting these reaction functions into (22), and then differentiating (22) with respect to $p_1$, we can obtain $p_1 = \frac{-2A - a0 + 2Ab - a0b - bc0 + b^2 c0 - c1 + bc1 - c2 + bc2}{4(-1+b)}$. Substituting $p_1 = \frac{-2A - a0 + 2Ab - a0b - bc0 + b^2 c0 - c1 + bc1 - c2 + bc2}{4(-1+b)}$ into the aforementioned $w_1 = \frac{a_0 + c_1 - bc_1 - p_1 + bp_1}{1-b}$ yields the equilibrium price.

## 4.8 Scenario EMF

In scenario EFM, the farmers not only sell biomass to the power plant through the middleman but also directly to the power plant; and the power plant provides electricity demand information to both the upstream farmers and middlemen. Therefore, in this scenario, the profit functions of each supply chain member are:

$$E(\pi_P|f) = p_1 d_1^E + p_2 d_2 - c_3(d_1^E + d_2) \tag{24}$$

$$E(\pi_M|f) = (p_1 - w_1 - c_2)d_1^E \tag{25}$$

$$E(\pi_F|f) = (w_1 - c_1)d_1^E + (p_2 - c_0)d_2 \tag{26}$$

By employing the reverse solution method: first, differentiate (26) with respect to $w_1$ and $p_2$, we obtain the reaction functions $w_1 = \frac{A + c_1 - bc_1 - p_1 + bp_1}{1-b}$ and $p_2 = \frac{A + c_0 - bc_0}{2(1-b)}$. Substituting these reaction functions into (25), and then differentiating (25) with respect to $p_1$, we can obtain $p_1 = \frac{-3A + Ab - bc_0 + b^2 c_0 - c_1 + bc_1 - c_2 + bc_2}{4(-1+b)}$. Substituting $p_1 = \frac{-3A + Ab - bc_0 + b^2 c_0 - c_1 + bc_1 - c_2 + bc_2}{4(-1+b)}$ into the aforementioned $w_1 = \frac{A + c_1 - bc_1 - p_1 + bp_1}{1-b}$ yields the equilibrium price.

Table 4 shows the equilibrium prices under four scenarios for farmers choosing channel encroachment.

Upon comparing the equilibrium prices, the following propositions are derived.

Proposition 1: For $w_1$ and $p_1$, when $A > a_0$, in single-channel $w_1^{NM} < w_1^{NN} < w_1^{NFM} < w_1^{NF}$, $p_1^{NM} < p_1^{NN} = p_1^{NF} < p_1^{NFM}$; in dual-channel $w_1^{EM} < w_1^{EN} < w_1^{EFM} < w_1^{EF}$, $p_1^{EM} < p_1^{EN} = p_1^{EF} < p_1^{EFM}$. Furthermore, $E(w_1^{Ni}) > E(w_1^{Ei})$, and $E(p_1^{Ni}) < E(p_1^{Ei})$, where $i = N/F/M/MF$.

**Table 4. Equilibrium pricing by scenario in the encroachment channel.**

|  | EN | EF | EM | EFM |
|---|---|---|---|---|
| $w_1$ | $\frac{1}{4}\left(\frac{a_0(1+b)}{1-b} - bc_0 + 3c_1 - c_2\right)$ | $\frac{4A + a_0(-3+b) + (-1+b)(bc_0 - 3c_1 + c_2)}{4(1-b)}$ | $\frac{1}{4}\left(-2A + \frac{a_0(3-b)}{1-b} - bc_0 + 3c_1 - c_2\right)$ | $\frac{1}{4}\left(\frac{A(1+b)}{1-b} - bc_0 + 3c_1 - c_2\right)$ |
| $p_1$ | $\frac{1}{4}\left(\frac{a_0(3-b)}{1-b} + bc_0 + c_1 + c_2\right)$ | $\frac{1}{4}\left(\frac{a_0(3-b)}{1-b} + bc_0 + c_1 + c_2\right)$ | $\frac{1}{4}\left(2A + \frac{a_0(1+b)}{1-b} + bc_0 + c_1 + c_2\right)$ | $\frac{1}{4}\left(\frac{A(3-b)}{1-b} + bc_0 + c_1 + c_2\right)$ |
| $p_2$ | $\frac{1}{2}\left(\frac{a_0}{1-b} + c_0\right)$ | $\frac{1}{2}\left(\frac{A}{1-b} + c_0\right)$ | $\frac{1}{2}\left(\frac{a_0}{1-b} + c_0\right)$ | $\frac{1}{2}\left(\frac{A}{1-b} + c_0\right)$ |

Proposition 1 illustrates the impact of biomass power plant demand information sharing and farmer channel encroachment on equilibrium prices. It is observed that regardless of farmer encroachment when the power plant only shares information with the middlemen, the wholesale price at which farmers sell biomass to the middlemen decreases; otherwise, the corresponding wholesale price $w_1$ increases. This is due to the advantage middlemen exploit when possessing demand information, which leads to a decrease in wholesale price. However, when farmers possess demand information, they adjust the wholesale price to ensure better profits. As for the unit selling price $p_1$ for the middlemen, it is noteworthy that regardless of farmer encroachment, when demand information is exclusively held by the middlemen, the unit price $p_1$ actually decreases. This is because when demand information forecasts high values ($A > a_0$), the middlemen are more willing to lower unit prices to gain more market share and maximize their profits. Additionally, when demand information forecasts low values ($A < a_0$), these results take on the opposite form.

Furthermore, it is found that given a certain encroachment scenario, the expected equilibrium price remains unchanged. Moreover, the choice of farmer channel encroachment reduces the expected wholesale price. This is because when farmers encroach, in order to maintain higher sales volume, farmers will reduce the wholesale price, as the unit selling price between the middleman and the power plant will correspondingly increase to ensure maximum profit for the middleman. This is known as the wholesale price effect [58].

## 5. Equilibrium comparison

The information sharing strategy for the power plant is determined using the reverse solution method.

Proposition 2: Given an encroachment strategy, for the farmer,
$$E(\pi_F^{NM}) < E(\pi_F^{NN}) < E(\pi_F^{NFM}) < E(\pi_F^{NF}); \ E(\pi_F^{EM}) < E(\pi_F^{EN}) < E(\pi_F^{EFM}) < E(\pi_F^{EF}).$$

Proposition 2 indicates that regardless of whether farmers encroach, they all prefer to exclusively access the electricity plant's demand information. This is because when the power plant shares demand information only with farmers, they can use this information to set more reasonable wholesale prices and maximize profits in competition. Additionally, we find that when the power plant shares information only with the middleman, farmers' expected profits decrease. This is because when the middleman has demand information, they are likely to set a more reasonable unit selling price $p_1$. In this case, farmers are forced to set a wholesale price lower than $p_1$ to get more sales. In other words, the middleman's use of information asymmetry harms farmers' profits. This conclusion aligns with reality. In practice, regardless of whether channel encroachment occurs, farmers should dismantle the information barriers inherent in traditional channels. They should proactively establish direct communication with biomass power plants via online platforms or on their own initiative. This approach ensures more effective information provision and thereby strengthens their decision-making process. To vividly illustrate this conclusion, We compare the equilibrium returns of farmers in different scenarios in Fig 3 (Similar to existing studies Wang et al. [50] and Liu et al. [56], set $a_0 = 15; c_1 = 2; c_2 = 1; c_3 = 0.8; c_0 = 2.5; v = 1; b = 0.6$).

Proposition 3: Given a specific encroachment strategy, for the middleman,
$$E(\pi_M^{NF}) < E(\pi_M^{NN}) < E(\pi_M^{NFM}) < E(\pi_M^{NM}); \ E(\pi_M^{EF}) < E(\pi_M^{EN}) < E(\pi_M^{EFM}) < E(\pi_M^{EM}).$$

Proposition 3 indicates that when the power plant exclusively shares demand information with the middleman, its profit is maximized. However, sharing demand information exclusively with the farmer will harm the middleman's profit. As shown in Fig 4, this conclusion is intuitive, so the explanation has been omitted. In reality, middlemen should strengthen cooperation with biomass power plants and actively seek more accurate demand information to

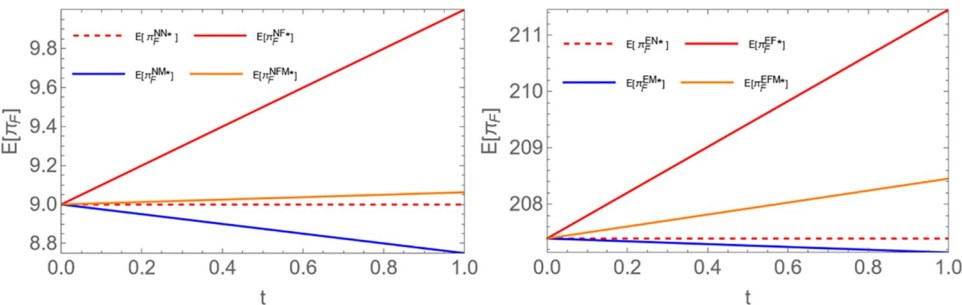

**Fig 3. Comparative chart of farmers' equilibrium returns.**

take advantage of the information gap and gain more benefits. Additionally, when there is an abundant supply of biomass raw materials, regardless of whether farmers conduct channel encroachment, power plants should give priority to providing middlemen with more accurate demand information to increase their own profits. (See the figure below; $a_0 = 15; c_1 = 2; c_2 = 1; c_3 = 0.8; c_0 = 2.5; v = 1; b = 0.6$).

Proposition 4: Given the non-encroachment scenario, $E(\pi_P^{NF}) = E(\pi_P^{NN}) < E(\pi_P^{NFM}) < E(\pi_P^{NM})$.

Proposition 4 suggests that in a single-channel supply chain structure, power plants should prioritize sharing demand information with the middleman. This is because when sharing information with the upstream middleman, the middleman can set more reasonable prices in response to the power plant's demand. Specifically, the middleman first sets the unit selling price of biomass raw materials $p_1$, then the farmer sets the wholesale price to meet the middleman's biomass raw material procurement demand. In addition, when power plants share demand information with both upstream participants, the vertical competition between farmers and middlemen intensifies, which not only affects the stability of supply but also exposes biomass power plants to cost pressures arising from upstream supply competition, ultimately leading to a decline in their profits. Finally, we find that when the power plant shares demand information only with the farmer, the expected profit of the power plant remains unchanged. In practice, within China, the supply model for biomass feedstocks, which is predominantly controlled by middlemen, continues to maintain a substantial presence. To guarantee the stability of feedstock supply, biomass power plants must forge long-term cooperative alliances with middlemen, facilitated by the sharing of demand information.

Proposition 5: Given the encroachment scenario, when b is less than 0.5, $E(\pi_P^{EN}) < E(\pi_P^{EF}) < E(\pi_P^{EM}) < E(\pi_P^{EFM})$; When $b$ is greater than 0.5, $E(\pi_P^{EF}) < E(\pi_P^{EN}) < E(\pi_P^{EM}) < E(\pi_P^{EFM})$.

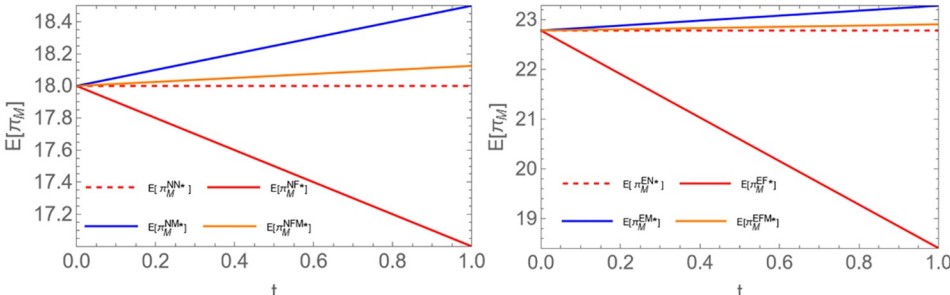

**Fig 4. Comparative chart of equilibrium returns of middlemen.**

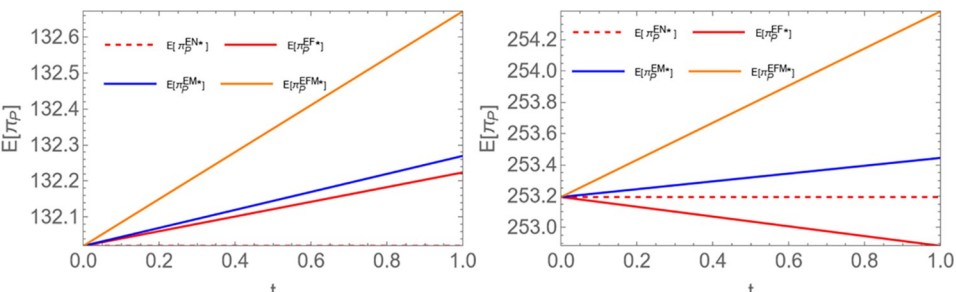

**Fig 5. Comparison of equilibrium returns from biomass power plants.**

Proposition 5 indicates that under normal circumstances, when the power plant shares demand information, upstream participants can better meet the needs of the power plant, leading to higher profits for the power plant. However, as shown in Fig 5, we observe that when the channel competition intensity is relatively high, sharing information only with farmers can harm the power plant's profits. This is because when farmers encroach the channel and the channel competition intensity is high, sharing information exclusively with farmers intensifies channel competition, resulting in price wars. Both parties may adopt aggressive strategies in their competition for market share, potentially leading to a price war that weakens the bargaining power of biomass power plants and subsequently results in a decline in profits. (See the figure below; $a_0 = 15$; $c_1 = 2$; $c_2 = 1$; $c_3 = 0.8$; $c_0 = 2.5$; $v = 1$. Among them, the left figure is $b = 0.3$, and the right is $b = 0.6$).

On the other hand, we observe that, unlike the single-channel scenario, in the encroachment scenario, the power plant always shares information with all upstream participants. This is because when all upstream entities have access to information, they can better utilize it, increasing the efficiency of the entire supply chain [59] to meet the demands of the terminal power plant. This conclusion indicates that in a dual-channel structure, the power plant should actively forecast future demand and share demand information fully with upstream participants.

Proposition 6: Farmers always prefer to encroach (the equilibrium outcome is EFM).

Proposition 6 indicates that in reality, farmers should strive to broaden their sales channels for biomass feedstocks. This is because, despite the potential reduction in revenue for the middleman channel due to the encroachment, the increased income from the encroached channel compensates for the loss in middleman channel profits, ultimately leading to overall gains. This is consistent with reality, as in some countries, an increasing number of farmers are forming associations, establishing resident committees, and adding direct channels to sell biomass raw materials [60].

## 6. Conclusion

This study focuses on the analysis of demand information sharing strategies of power plants and the interaction of these strategies with other members of the biomass supply chain under a competitive behavior encroachment scenario, which is an area that has yet to be fully explored in previous literature. We constructed a biomass supply chain network consisting of farmers (F), middlemen (M), and power plants (P). In this network, farmers are responsible for collecting biomass and wholesale it to middlemen, who then sell it to power plants for electricity generation. In some cases, it may be possible for farmers to sell biomass to power plants through direct sales channels, bypassing middlemen. Power plants have data to predict future demand

for biomass and need to decide whether and how to share this demand information with upstream supply chain members. By building an analytical model, we explored the information sharing mechanism and analyzed the potential impact of each party's strategic choices on supply chain performance. Through this study, we aim to provide a deeper understanding of the role of information sharing in improving the efficiency of biomass supply chain management.

## 6.1 Major findings

This study analyzes eight different strategic scenarios in the biomass power supply chain and draws the following main conclusions. First, the equilibrium price of each member of the biomass power supply chain is significantly affected by information sharing strategies and market expropriation behaviors. Second, we observe that both middlemen and farmers tend to monopolize power plant biomass demand information when faced with market cannibalization. When adversaries have exclusive access to this information, their profits are negatively affected. Both middlemen and farmers in the supply chain seek information advantages to enhance their market position. Contrasting the theoretical frameworks established in the current literature [61], our proposition posits that the excessive sharing of demand information by biomass power plants may paradoxically lead to negative outcomes, ultimately eroding their profitability. Biomass power plants must exercise caution in selecting their information sharing strategies, taking into account the intensity of competition within their respective supply channels. Consequently, our paper introduces a fresh perspective to the corpus of research exploring information sharing strategies in the context of biomass power supply, providing managers who must decide on information sharing strategies with practical insights and recommendations. Finally, our research finds that it is often advantageous for farmers to adopt a market encroachment strategy because the increased income from the encroached channel makes up for the profit loss from the intermediate channel. This finding is concordant with the conclusions drawn from existing studies, which indicate that in reality, farmers, either spontaneously or under government guidance, have formed various formal organizations and established direct channels for selling biomass to power plants [62].

## 6.2 Managerial implications

This study has certain implications for the behavioral decision-making of members of the biomass supply chain. Different from the traditional view, this paper presents the optimal power plant demand information sharing strategy under information asymmetry and how biomass power plants can disclose information to maximize profits when biomass feedstock supplier farmers choose to encroach or not. This will significantly improve the coordination and stability of the biomass power supply chain.

Specifically, (1) From the perspective of farmers, regardless of whether power plants share demand information, farmers should strive to expand their biomass sales channels as much as possible to enhance their own profits. At the same time, actively seeking relevant information through both online and offline channels is a wise choice. (2) For middlemen, it is essential to actively pursue long-term cooperation with biomass power plants through various means in order to acquire demand information more rapidly and accurately, thus gaining a proactive position in the competition. In reality, middlemen should more appropriately stabilize their cooperation with power plants by adjusting their own pricing decisions, thereby reducing the intensity of channel competition. (3) The demand information sharing strategy of biomass power plants can be affected by encroachment scenarios. When farmers do not choose to encroach, it is recommended that power plants share demand information exclusively with

middlemen to ensure the stability of biomass feedstock supply; when there is an encroachment, it is recommended that power plants share information with all upstream players as much as possible in order to broaden the feedstock collection channels. At the same time, when the competition intensity of the encroachment channel is high, it is recommended that the power plants avoid sharing the demand information with the farmers exclusively, which may cause unnecessary price wars.

In addition to the specific members of the biomass power generation supply chain, this study also holds certain implications for government entities. As functional institutions, governments should strengthen the establishment of biomass information sharing platforms by publishing market information, collecting supply and demand data, and other means to provide support for decision-making among supply chain members. Furthermore, the government should formulate laws, regulations, and industry standards to regulate the order of the biomass market and promote fair competition. Additionally, the government can guide and encourage farmers to establish cooperatives for direct or indirect sales of biomass raw materials through incentive measures such as tax subsidies, thereby enhancing farmers' income and promoting the stable development of the biomass power generation supply chain.

### 6.3 Future researches

This study demonstrates several important limitations while providing insights into biomass power supply chain strategies. First, we assume that the power plant's cost of obtaining biomass demand information is zero. Indeed, the costs of forecasting technology, equipment, and personnel may significantly affect equilibrium outcomes. Future research could include these costs to assess their specific impact on strategy choice and equilibrium outcomes. Second, the costs to farmers of collecting biomass feedstock may change over time, and changes in these costs may have implications for prices and strategies in the supply chain. Future research could explore the impact of cost dynamics on equilibrium outcomes. Finally, this study only considers the scenario of a single middleman. In reality, there may be multiple middlemen competing for biomass feedstock, which may lead to more complex market structures and competitive strategies. Therefore, considering the situation of competition among multiple middlemen will be an important direction for our future research.

## Supporting information

**S1 File. Appendix 1 derivation process.**
(PDF)

## Author Contributions

**Conceptualization:** Xin Wu.

**Data curation:** Xin Wu, Peng Liu.

**Formal analysis:** Xin Wu.

**Funding acquisition:** Guangyin Xu.

**Investigation:** Xin Wu, Peng Liu, Jing Gao.

**Methodology:** Xin Wu, Jin Li, Jing Gao, Guangyin Xu.

**Software:** Guangyin Xu.

**Visualization:** Xin Wu, Peng Liu.

**Writing – original draft:** Xin Wu.

**Writing – review & editing:** Xin Wu, Peng Liu, Guangyin Xu.

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
