## [Decision Letter · Decision Letter 0]

28 Mar 2024

PONE-D-24-05811Information Sharing And Channel Encroachment In Biomass Supply ChainsPLOS ONE

Dear Dr. Xu,

Thank you for submitting your manuscript to PLOS ONE. After careful consideration, we feel that it has merit but does not fully meet PLOS ONE’s publication criteria as it currently stands. Therefore, we invite you to submit a revised version of the manuscript that addresses the points raised during the review process.

We look forward to receiving your revised manuscript.

Kind regards,

Fuli Zhou

Academic Editor

PLOS ONE

3. PLOS requires an ORCID iD for the corresponding author in Editorial Manager on papers submitted after December 6th, 2016. Please ensure that you have an ORCID iD and that it is validated in Editorial Manager. To do this, go to ‘Update my Information’ (in the upper left-hand corner of the main menu), and click on the Fetch/Validate link next to the ORCID field. This will take you to the ORCID site and allow you to create a new iD or authenticate a pre-existing iD in Editorial Manager. Please see the following video for instructions on linking an ORCID iD to your Editorial Manager account: https://www.youtube.com/watch?v=_xcclfuvtxQ".

[This study was supported by Henan province agricultural science and technology research project (Grant No: 192102110205).]

 [The author(s) received no specific funding for this work.]

5. We note that your Data Availability Statement is currently as follows: [All relevant data are within the manuscript and its Supporting Information files.]

Additional Editor Comments:

The abstract section should be revised, and the research gap should be also addressed in the first section.

Reviewers' comments:

Reviewer's Responses to Questions

**Comments to the Author**

1. Is the manuscript technically sound, and do the data support the conclusions?

Reviewer #1: Yes

Reviewer #2: Yes

2. Has the statistical analysis been performed appropriately and rigorously? 

Reviewer #1: Yes

Reviewer #2: Yes

3. Have the authors made all data underlying the findings in their manuscript fully available?

Reviewer #1: Yes

Reviewer #2: Yes

4. Is the manuscript presented in an intelligible fashion and written in standard English?

Reviewer #1: Yes

Reviewer #2: Yes

5. Review Comments to the Author

Reviewer #1: This paper investigates the interactions between the information sharing strategy and channel encroachment in the straw power supply chain. I Think this paper is interesting, but several problems still need to be addressed. The detailed comments are as follows:

(1) The first part introduces the background and motivation of the research, and the general idea is good. It would be more helpful to offer more examples, especially information sharing.

(2) The second part reviews the literature of the three related fields. Authors are also advised to increase literature citations about PLOS ONE journal. It is suggested to list the relevant literature and contributions as well as the innovation of this paper in the form of table.

(3) In Section 3, I recommend adding the descriptions to explain why this decision sequence for the game is adopted.

(4) There are some areas of expression that need to be improved. For example, in the first line of the second paragraph of part 3, I don't understand what function is the 'existing function' here? Where does it come from? Why is it used? What's the difference between this pre-existing function in the original literature and in my current paper?

(5) The paper should explain more about the mechanism behind the conclusions. Specifically, the theoretical analysis is simple and insufficient, and the reason behind some key results is not clear. For example, Corollaries 4 shows that sharing information of biomass power plants will lead to vertical competition of raw material supply channels. I would like to see a more specific description after the 'vertical competition', for example, the price war caused by vertical competition leads to the reduction of profits of biomass power plants? Other analyses have the same problem.

(6) I also recommend adding several management insights in Sections 4 and 5.

Reviewer #2: The overall logic of this paper is clear, and the game theory model is used to explore the optimal information sharing strategy for farmers to choose whether to occupy the biomass feedstock supply of biomass power plants. Through the detailed analysis of 8 different information sharing scenarios, a more powerful research conclusion is obtained.In addition, the structure of the paper is reasonable, and the selected research questions are interesting.For example, the research on the sustainable development of the biomass raw material supply chain is a very important research hotspot in China's current development and has strong research necessity. This paper chooses a good research entry point. Personally, I think it is a good research paper. However, there are still some shortcomings that need to be further improved to better meet the standards of public publication.

1.Please adjust the expression of the first sentence of the second paragraph of the introduction. The second paragraph mainly expresses the view that "biomass power generation still has considerable space for development", which has no transitional relationship with the previous paragraph, it is not recommended to use "HOWEVER..." Sentence patterns to express ideas.

2.It is suggested to add "theoretical contributions" to the "6 Conclusions" section. This part should respond to the research gap proposed above through literature dialogue, in order to highlight the theoretical contribution of this research.

3.This research question has strong application orientation, and it is suggested to further enrich the content of "Managerial implications". Specific improvement measures can be proposed from the perspective of different relevant subjects, such as how the government should use the conclusions of this study to improve relevant work and so on.

6. PLOS authors have the option to publish the peer review history of their article (what does this mean?). If published, this will include your full peer review and any attached files.

Reviewer #1: No

Reviewer #2: No

---

## [Decision Letter · Decision Letter 1]

20 Aug 2024

PONE-D-24-05811R1Information Sharing And Channel Encroachment In Biomass Supply ChainsPLOS ONE

Dear Dr. Xu,

Thank you for submitting your manuscript to PLOS ONE. After careful consideration, we feel that it has merit but does not fully meet PLOS ONE’s publication criteria as it currently stands. Therefore, we invite you to submit a revised version of the manuscript that addresses the points raised during the review process.

We look forward to receiving your revised manuscript.

Kind regards,

Tengfei Nie

Academic Editor

PLOS ONE

Journal Requirements:

**Additional Editor Comments:**

The authors need to make some further minor revisions.

Reviewers' comments:

Reviewer's Responses to Questions

**Comments to the Author**

1. If the authors have adequately addressed your comments raised in a previous round of review and you feel that this manuscript is now acceptable for publication, you may indicate that here to bypass the “Comments to the Author” section, enter your conflict of interest statement in the “Confidential to Editor” section, and submit your "Accept" recommendation.

Reviewer #1: All comments have been addressed

Reviewer #3: (No Response)

2. Is the manuscript technically sound, and do the data support the conclusions?

Reviewer #1: Yes

Reviewer #3: Yes

3. Has the statistical analysis been performed appropriately and rigorously? 

Reviewer #1: Yes

Reviewer #3: Yes

4. Have the authors made all data underlying the findings in their manuscript fully available?

Reviewer #1: Yes

Reviewer #3: Yes

5. Is the manuscript presented in an intelligible fashion and written in standard English?

Reviewer #1: Yes

Reviewer #3: Yes

6. Review Comments to the Author

Reviewer #1: The author carefully revised the paper and responded to the opinions of the reviewers.

Meanwhile, the format of the paper also meets the requirements of the journal.

From these aspects, it can be seen that the author takes this paper very seriously.

I suggest that this paper can be accepted.

Reviewer #3: Dear authors,

I now finished reading the manuscript of “Information Sharing And Channel Encroachment In Biomass Supply Chains”. This research the interactions between the information sharing strategy and channel encroachment in the straw power supply chain. I Think this paper is interesting, but several problems still need to be addressed.

1. In section3, there are two cases of whether farmers invade or not in d1, which should be further distinguished by using corner label. In addition, it is suggested that the decision sequence can be drawn in the form of a chart and then attached to explain, which is easy for readers to read and understand.

2. Table3 and 4 summarize the pricing conclusions in all cases, I suggest that the position should be adjusted, not in the case of 4.1Scenario NN.

3. The numerical analysis in Section 5. Will the change of the numerical value affect the conclusion? What are the values based on?

4. The analysis of proposition should also add the connection with reality, and put forward the corresponding management implications combined with the conclusion.

7. PLOS authors have the option to publish the peer review history of their article (what does this mean?). If published, this will include your full peer review and any attached files.

Reviewer #1: No

Reviewer #3: No

---

## [Author Response · Author response to Decision Letter 1]

30 Aug 2024

Please refer to Annex “Response to Reviewers” for details.

---

## [Editor Report · Decision Letter 2]

4 Sep 2024

Information Sharing And Channel Encroachment In Biomass Supply Chains

PONE-D-24-05811R2

Dear Dr. Xu,

We’re pleased to inform you that your manuscript has been judged scientifically suitable for publication and will be formally accepted for publication once it meets all outstanding technical requirements.

Kind regards,

Tengfei Nie

Academic Editor

PLOS ONE
---

## [Editor Report · Acceptance letter]

11 Sep 2024

PONE-D-24-05811R2 

PLOS ONE

Dear Dr. Xu, 

I'm pleased to inform you that your manuscript has been deemed suitable for publication in PLOS ONE. Congratulations! Your manuscript is now being handed over to our production team.

Kind regards, 

on behalf of

Dr. Tengfei Nie 

Academic Editor

PLOS ONE